# Strengthening of Continuous Reinforced Concrete Deep Beams with Large Openings Using CFRP Strips

**DOI:** 10.3390/ma14113119

**Published:** 2021-06-06

**Authors:** Mohammed Riyadh Khalaf, Ali Hussein Ali Al-Ahmed, Abbas AbdulMajeed Allawi, Ayman El-Zohairy

**Affiliations:** 1Department of Civil Engineering, University of Baghdad, Baghdad 17001, Iraq; m.khalaf1101@coeng.uobaghdad.edu.iq (M.R.K.); dr.ali-alahmed@coeng.uobaghdad.edu.iq (A.H.A.A.-A.); A.Allawi@uobaghdad.edu.iq (A.A.A.); 2Department of Engineering and Technology, Texas A&M University-Commerce, Commerce, TX 75429, USA

**Keywords:** reinforced concrete deep beams, large openings, strengthening, CFRP strips, mode of failure, finite element, parametric study

## Abstract

To accommodate utilities in buildings, different sizes of openings are provided in the web of reinforced concrete deep beams, which cause reductions in the beam strength and stiffness. This paper aims to investigate experimentally and numerically the effectiveness of using carbon fiber reinforced polymer (CFRP) strips, as a strengthening technique, to externally strengthen reinforced concrete continuous deep beams (RCCDBs) with large openings. The experimental work included testing three RCCDBs under five-point bending. A reference specimen was prepared without openings to explore the reductions in strength and stiffness after providing large openings. Openings were created symmetrically at the center of spans of the other specimens to represent 40% of the overall beam depth. Moreover, finite elements (FE) analysis was validated using the experimental results to conduct a parametric study on RCCDBs strengthened with CFRP strips. The results confirmed reductions in the ultimate load by 21% and 7% for the un-strengthened and strengthened specimens, respectively, due to the large openings. Although the large openings caused reductions in capacities, the CFRP strips limited the deterioration by enhancing the specimen capacity by 17% relative to the un-strengthened one.

## 1. Introduction

Deep beams are defined as beams with a clear span (L_c_) to overall depth (H) ratio that is less than or equal to four [1]. In the structural engineering industry, reinforced concrete continuous deep beams (RCCDBs) are a subject of great interest. Transfer girders, pile caps, tanks, folded frames, and foundation walls are some examples of RCCDBs that carry several small loads and pass them to a small number of reaction points [2,3]. Figure 1 illustrates RCCDBs with high span/depth ratios.

Openings in the web of RCCDBs are commonly used to facilitate important utilities, such as air conditioner ducts and electricity cables, and to provide functionality, such as doors and windows [2]. These openings can be defined as small or large openings based on the ratio of the web opening depth (h_o_) to the total depth of the section (H). If this ratio is less than 25%, the opening might be considered a small opening [4]. Providing openings could result in significant reductions in strength and stiffness, which could also lead to significant safety risks. Therefore, strengthening techniques can be used to recover and regain the beam strength due to openings. External strengthening using CFRP strips is considered an appropriate strengthening technique [5]. The greatest benefits of using CFRP strips are lightweight, high-strength, speed, ease of application, and the ability to form different shapes on-site.

Previous research studies have been conducted on the behavior and strengthening of simply supported reinforced concrete deep beams with and without openings [6,7,8,9,10,11,12,13]. The creation of rectangular and circular openings in shear spans of the reinforced concrete deep beams affect the load-carrying capacity, where the reduction of the failure load reached 66% [6,8]. The ultimate capacity of simply supported reinforced concrete deep beams was not influenced by the opening if the depth of the top chord was more than or equal to the depth of the concrete stress block at the ultimate state when the openings were in the pure flexure zone [9]. Symmetrical, unsymmetrical, and centered web openings were investigated through load-deflection curves, crack patterns, and absorbed energy [13]. Vertical and inclined U-wrapped schemes were used to strengthen the deep beams around the openings [7,8,9,10]. Using inclined strips of CFRP was more effective in upgrading the stiffness and shear strength of deep beams [8,10]. The addition of horizontal CFRP strips to the system of vertical CFRP strips added improvement in the ultimate strength [10]. Moreover, the most effective number of CFRP layers was investigated for reinforced concrete simply supported deep beams with large openings in the shear zone. Two and three layers were the most effective for opening sizes of 150 and 200 mm, respectively [11].

An advanced nonlinear FE model was used to understand the behavior of deep beams retrofitted with fiber-reinforced polymer (FRP) sheets [12]. The crack formation and propagation in epoxy were predicted by incorporating a moving mesh modeling. The debonding mechanism was modeled using interface elements [14]. The CFRP sheets were modeled using Shell elements with multilayers including the adhesive layer. This adhesive layer was considered the bond between the CFRP sheets and concrete [15]. Increasing numbers of CFRP layers effectively enhanced the load-carrying capacity [11].

On the other hand, different strengthening techniques were investigated for the RCCDBs. These techniques included steel reinforced grout, a fabric-reinforced cementitious matrix, and external pre-stressed strands [16,17,18]. The bond of the steel-reinforced grout system was influenced by the curing time and the ultimate capacity was linearly increased with the age of the composite [16]. The shear strength was improved by using the fabric-reinforced cementitious matrix but not proportionally to the number of fabric plies installed [17]. However, the physical properties of using CFRP laminates provided an effective choice as a strengthening technique [5].

Most of the previous studies focused on strengthening of simply supported reinforced concrete deep beams and there are limited reported works (up to date) related to the application of CFRP strips, as a strengthening technique, for RCCDBs with large openings. Therefore, this paper aims to investigate experimentally and numerically the effectiveness of using carbon fiber reinforced polymer (CFRP) strips to externally strengthen reinforced concrete continuous deep beams (RCCDBs) with large openings. The experimental work included testing three RCCDBs under five-point bending. A reference specimen was prepared without openings to explore the reductions in strength and stiffness after providing large openings. Openings were created symmetrically at the center of spans of the other specimens to represent 40% of the overall beam depth. Moreover, finite elements (FE) analysis was validated using the experimental results to conduct a parametric study on RCCDBs strengthened with CFRP strips.

## 2. Experimental Program

The experimental work included testing three RCCDBs under five-point bending (see Table 1). A reference specimen was prepared without openings to explore the reductions in strength and stiffness after providing large openings.

### 2.1. Description and Details of the Tested Specimens

The tested specimens had a total length of 2500 mm, a clear span of 1190 mm, a cross-section of 160 mm × 400 mm, and a shear span of 595 mm (see Figure 2). The first specimen (CDB–Solid) was prepared without an opening (as a reference specimen). Openings of dimensions 160 mm × 160 mm were created in the second and third specimens (CDB–O–U and CDB–O–S). These openings were created symmetrically at the center of spans of the other specimens to represent 40% of the overall beam depth. Steel plates 160 mm × 70 mm × 10 mm were used at locations of the applied loads and supports to avoid any stress concentration and premature failure in concrete.

The arrangement of reinforcements consisted of 3φ12 mm rebars as longitudinal bottom reinforcement and 2φ12 mm rebars as top reinforcement. In addition, 2φ12 mm with a length of 700 mm was provided at the hogging moment region over the mid-support. Stirrups of φ6 mm with a spacing of 70 mm were used as shear reinforcement. Shrinkage reinforcement of 10φ6 mm was used, as shown in Figure 3.

Unidirectional CFRP strips, Pro-fiber CW 450 (DCP), were used in the strengthening process with an effective thickness of 0.255 mm. The strengthening scheme is shown in Figure 4. The CFRP strips were applied around the openings as horizontal, vertical, and diagonal stripes. This strengthening scheme is the most effective one for strengthening openings with CFRP strips to oppose the diagonal cracking during loading [8].

### 2.2. Material Properties

Normal weight concrete with a cylindrical compressive strength of 23 MPa was produced for casting the tested specimens. The yield stress and ultimate strength of steel reinforcement were 510 and 625 MPa, respectively, for bars with a diameter of 6 mm and 650 and 730 MPa for bars with a diameter of 12 mm. The mechanical properties of the CFRP strips are provided in Table 2. In addition, a two-component adhesive, Quickmast 350 (Impregnating Resin (DCP)), was used to bond the laminates to concrete. These components were mixed by a ratio of 2 to 1 of resin to hardener by weight. This thin layer of epoxy was modeled as a 1.0 mm thin layer between the CFRP laminates and concrete. The mechanical properties of the used epoxy are listed in Table 3.

### 2.3. Instrumentation and Test Setup

A hydraulic jack (INSTRON, Norwood, MA, USA) with a capacity of 1000 kN was used to perform the laboratory tests. The static testing was performed using a single hydraulic jack and then the load was divided into two concentrated loads on each span by using a spreader loading beam (see Figure 5). The applied load was measured using a load cell connected to the data acquisition system. Displacement measurements were recorded at the midpoints of the bottom surface of each specimen using linear variable displacement transducers (LVDTs).

## 3. Test Results and Discussions

To monitor the static behaviors of the tested specimens, initial cracks, modes of failure, ultimate capacities and deflections, and load-deflection relationships were observed. Table 4 summarizes the experimental results for the tested specimens.

### 3.1. Initial Cracks and Modes of Failure

For specimen CDB–Solid, the initial crack was formed at the center of each span as flexural cracks. Near supports, diagonal cracks (shear cracks) appeared and propagated to the loading points until concrete crushing occurred at both point loads and supports, as shown in Figure 6a. Shear failure was the mode of failure of the reference specimen CDB–Solid. The initial crack was observed at corners of the openings of specimen CDB–O–U and propagated towards the loading and supporting points. The load path had been disturbed due to the existence of openings. The initial shear crack occurred in the corner of these openings. The reduction in the initial crack load was 30% relative to the reference specimen (CDB–Solid). Bearing failure at the top chords of openings was the mode of failure for the specimen with openings (CDB–O–U), as illustrated in Figure 6b. On the other hand, the addition of CFRP turned the load away from the openings. The diagonal CFRP strips were used to withstand the stress concentration at the corners of these openings and subsequently prevented the formation of the shear cracks at these corners. The initial crack load was increased by 50% relative to the un-strengthened specimen (CDB–O–U). As the load increased, concrete at the upper chord of one of the openings, see Figure 6c, was crashed with debonding of the CFRP strips.

### 3.2. Load Mid-Span Deflection Relationships

Figure 7 illustrates the load-deflection relationships at the centers of the spans of the tested specimens. The presence of large openings led to reductions in the stiffness, ultimate capacity, and ultimate deflection relative to specimen CDB–Solid. The presence of a large opening led to the loss of stiffness by 35% for specimen CDB–O–U. However, adding CFRP strips around these openings limited this reduction to 20% only, which means there was a 15% improvement relative to the un-strengthened one. Moreover, these openings decreased the ultimate capacities by 21% and 7% for specimens CDB–O–U and CDB–O–S, respectively. On the other hand, when specimen CDB–O–S was compared to specimen CDB–O–U, the percentage increase in the ultimate capacity was 17%. The CFRP strips helped to relieve stress concentrations at corners of the openings and subsequently improved the behavior of the strengthened beam.

## 4. Finite Element Analysis

The geometric and material non-linearity simulation of RCCDBs with and without openings was presented by proposing a FE model, using ABAQUS [19]. The proposed model was validated using the test findings presented in this paper. Moreover, an extensive parametric study was carried out to represent an important benchmark to predict the full static resistance of RCCDBs with openings under the effect of different parameters.

### 4.1. Constitutive Models of Materials

The concrete damage plasticity (CDP) model, which is available in ABAQUS, was used to simulate the concrete behavior [19]. This model requires the concrete compressive and tensile composition relation, damage parameters for cracking and crushing and other parameters of material such as dilation (φ), eccentricity (ε), compressive strength to uniaxial pressure ratio biaxial ($f_{bo}/f_{co}$), coefficient K, and viscosity parameters (μ) [20]. Table 5 lists the different parameters that were established from previous analyses and implemented in this study. The stress–strain relationship in compression that was proposed by Saenz [21] was adopted in this research and is illustrated in Figure 8a. On the other hand, the stress–strain relationship of concrete in tension was determined by using the exponential function proposed by Belarbi and Hsu [22] and is illustrated in Figure 8b.

The bilinear model, which is adopted in ABAQUS, was used to simulate the behavior of steel reinforcement, shrinkage reinforcement, and stirrups. For the linear isotropic part, it is defined by the modulus of elasticity of the reinforcement and the Poisson’s ratio, which are taken as 200 × 10^3^ MPa and 0.3, respectively, and away from this point, it plasticized perfectly, which was defined by the yield stress f_y_. However, a linear model was used to represent the stress–strain relationship for the CFRP laminates. This relationship was defined by the modulus of elasticity and tensile strength, which were provided by the manufacturer. The CFRP composites were treated as an orthotropic material by considering the direction of fibers as the main direction. The mechanical properties used in the FE simulation are listed in Table 2.

### 4.2. Element Type and Meshing Scheme

Concrete was idealized by using a solid element, C3D8, which has the capability of cracking in tension and crushing in compression. The steel rebars are simulated by using a three-dimensional truss element (T3D2). A 4-node shell element (S4) was used to simulate the CFRP strips. The used plates at supports and loading points were modeled using the solid finite element in ABAQUS. Many trials with different mesh sizes were carried out to improve the FE and avoid the in-convergence issues. Finally, cubes with maximum dimensions of 30 mm × 30 mm × 30 mm for concrete and squares of 30 mm × 30 mm for the CFRP strips were used in the mesh of the analyzed beams. The mesh geometry, loading arrangement of the FE model, CFRP strips, and details of the steel reinforcement are presented in Figure 9.

### 4.3. Boundary Conditions

The boundary conditions were modeled as a contentious beam, which was the same as those of the test setup. At one of the supports, the translation degrees of freedom were constrained in X and Y directions, which represented hinged support. In contrast, the translational degrees of freedom were constrained only in the Y direction for the other two supports, which represented roller supports.

The steel reinforcements were embedded inside the solid concrete elements with a perfect bond connection. Moreover, the full bond connection was assumed between the two surfaces of concrete and CFRP strips. The contact between the concrete surfaces and steel plates was assigned as tie constraints. The translational and rotational motion, as well as all other active degrees of freedom, were equal.

## 5. Validation of the FE Model

The experimental results were used to verify a reliable FE model able to simulate the static response of RCCDBs with large openings.

### 5.1. Load–Deflection Relationships

Figure 10, as well as Table 6, presents comparisons between the experimental and FE results in terms of deformations of the tested specimens. During the elastic stage, the response of the specimen from the two approaches was very close to each other. When the applied load reached the yielding load, the FE results became slightly stiffer. This difference in response was attributed to the assumed full contact between concrete and the strengthening strips as well as reinforcement. However, good agreement was obtained in general, and therefore, the validated FE model was used to carry out a parametric study.

### 5.2. FE Crack Pattern

The crack patterns of some specimens obtained from FE results in comparison with experimental cracks are shown in Figure 11. The FE crack patterns were represented through the option used to define post-cracking damage (stiffness degradation) properties for the concrete damaged plasticity material model. It can be visualized from Figure 11 that the FE crack patterns are in close agreement with the experimental cracks proving that the FE software ABAQUS accurately predicted the behavior of RCCDBs with large openings as well as using CFRP strips in strengthening.

## 6. Parametric Study

### 6.1. Effect of the Ratio of the Opening Dimensions

The ratio of the opening dimensions is defined as the length of the opening divided by the depth of the opening. In this parametric study, the ratios of the opening dimensions were chosen to be 1.0, 1.5, and 2.0 (i.e., opening length = 160, 240, and 320 mm). The opening depth was kept constant at 160 mm for comparison purposes. Each model was analyzed with and without CFRP strips to explore the effect of adding this strengthening technique on the overall behavior of RCCDBs with openings of different dimensions. Descriptions and details of the newly analyzed beams with opening ratios 1.5 and 2.0 are shown in Figure 12.

Figure 13 shows the effect of using openings with different dimensions ratios on the behavior of strengthened and un-strengthened RCCDBs relative to the solid beam (CDB–Solid). It was clear that as the opening ratio increased, the ultimate capacity was reduced, and the mid-span deflection was increased. Moreover, Table 7 summarizes these effects on the ultimate capacities and mid-span deflections. The percentages of improvements in the beam capacity and deflection for the strengthened beams were calculated corresponding to the un-strengthened one with the same opening ratio. The ratio of the opening dimensions affected the effectiveness of using CFRP strips as a strengthening technique for RCCDBs with openings. The ultimate capacity and mid-span deflection of the strengthened beam were improved as the ratio of the opening dimensions increased relative to the un-strengthened one (see Figure 14). The ultimate loads were increased by 18%, 23%, and 35% for beams with an opening ratio of 1.0, 1.5, and 2.0, respectively. In contrast, the mid-span deflections were decreased by 7%, 23%, and 40% for beams with ratios of the opening dimensions of 1.0, 1.5, and 2.0, respectively.

Figure 15 and Figure 16 show the effect of the ratios of the opening dimensions on the FE crack pattern of un-strengthened and strengthened beams. It was obvious that as the ratio of the opening dimensions increased, the concrete damage increased around the perimeter of the openings. Therefore, the effectiveness of using CFRP strips as a strengthening technique around these openings increased.

### 6.2. Effect of Load Distribution Factor

Another parameter was adopted in this section, which concerns the effect of load distribution factor (k) on the behavior of un-strengthened and strengthened RCCDBs with large openings. The load distribution factor (k) refers to the ratio of a single-span applied load to the total load (see Figure 17). In the present study, this factor was set as 0.25, 0.30, and 0.40, as listed in Table 8.

Figure 18, Figure 19, Figure 20 show the effect of the load distribution factor (k) on the load–deflection relationship of the analyzed beams. The data presented in these figures were for the applied load and mid-span deflection for the critical span (i.e., the span under the applied load of (1-k) of the total load). From these figures, it can be noticed that the decreasing of load distribution factor from 0.4 to 0.25 led to reductions in the ultimate capacity and an increase in the ultimate deflection compared to the beam with an equal load distribution factor (i.e., k = 0.5). Table 8 summarizes ultimate capacities and mid-span ultimate deflections for the analyzed beams.

For the strengthened beams with an opening ratio of 1.0, the reductions in the ultimate capacity were 18%, 32%, and 36% for load distribution factors of 0.4, 0.3, and 0.25, respectively, relative to the reference beam for this category (CDB–O–S–R1.0 with k = 0.5). In contrast, the percentage increase in mid-span ultimate deflections of these beams were 117%, 312%, and 371%, respectively.

For the strengthened beams with an opening ratio of 1.5, the reductions in the ultimate capacities were 18%, 31%, and 36% for load distribution factors of 0.4, 0.3, and 0.25, respectively, relative to the reference beam of this category (CDB–O–S–R1.5 with k = 0.5). However, the percentage increase in mid-span ultimate deflections of these beams were 62%, 171%, and 245%, respectively.

For the strengthened beams with an opening ratio of 2.0, the reductions in the ultimate capacity were 19%, 31%, and 36% for load distribution factors of 0.4, 0.3, and 0.25, respectively, relative to the reference beam of this category (CDB–O–S–R2.0 with k = 0.5). In contrast, the percentage increase in mid-span ultimate deflections of these beams were 46%, 121%, and 179%, respectively.

From these results, it can be noticed that the decreasing of load distribution factor from 0.4 to 0.25 led to the same trend of reduction in the ultimate capacities and increase in the ultimate deflections compared to the beams with an equal load distribution factor (i.e., k = 0.5).

## 7. Conclusions

The effectiveness of using CFRP strips was investigated experimentally and numerically to externally strengthen RCCDBs with large openings. The experimental work included testing three RCCDBs under five-point bending. Moreover, FE analysis was validated using the experimental results to conduct a parametric study on RCCDBs strengthened with CFRP strips. The following conclusions can be drawn from this research work.

1. Using CFRP strips restored part of the reduced capacity by 17% and decreased the mid-span ultimate deflection by 10% relative to the un-strengthened beam.

2. It was observed that using CFRP strips led to an increase in the first crack load by 5% and 50% relative to the solid and un-strengthened specimens, respectively.

3. The ratio of the opening dimensions affected the effectiveness of using CFRP strips as a strengthening technique for RCCDBs with openings. The ultimate capacity and mid-span deflection of the strengthened beam were improved as the opening ratio increased relative to the un-strengthened one.

4. The existence of CFRP strips had a minor effect on controling the behavior of RCCDBs with different load distribution factors. Reductions in the ultimate capacity and increase in the mid-span ultimate deflection were obtained as the load distribution factor decreased from 0.5 to 0.25.

5. The presence of large openings in RCCBDs led to reductions in the ultimate capacity by about 21% and initial stiffness by about 35%, and an increase in the mid-span ultimate deflection by about 21% was obtained compared to the solid beam.

## Figures and Tables

**Figure 1 materials-14-03119-f001:**
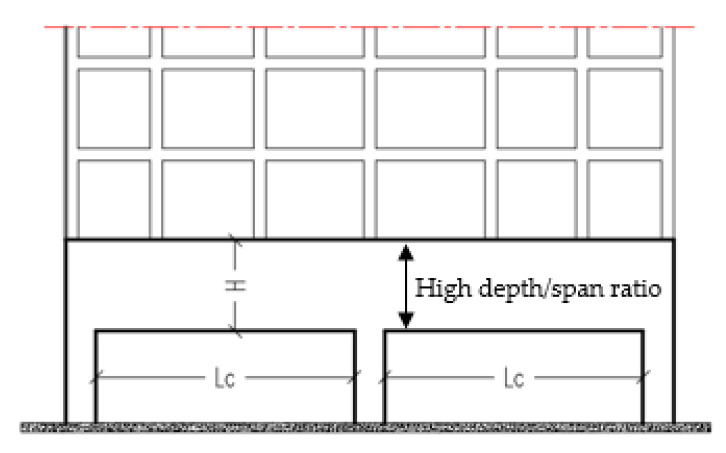
Illustration of a reinforced concrete continuous deep beam.

**Figure 2 materials-14-03119-f002:**
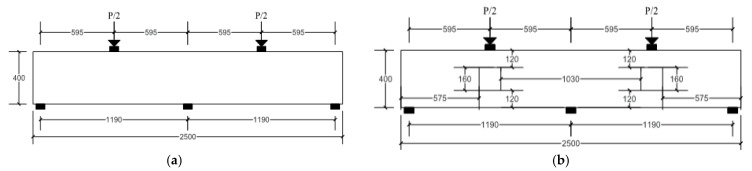
Typical layouts of the tested specimens (**a**) RCCDBs without openings; (**b**) RCCDBs with openings (all dimensions are in mm).

**Figure 3 materials-14-03119-f003:**
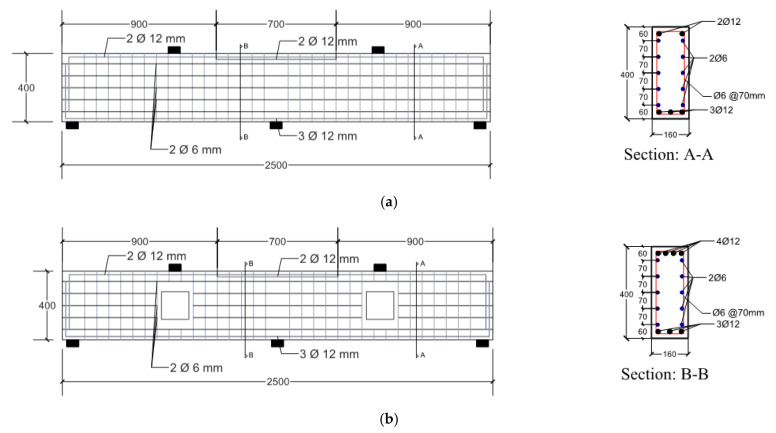
Details of steel reinforcement for the tested specimens: (**a**) specimen CDB–Solid; (**b**) specimens CDB–O–U and CDB–O–S (all dimensions are in mm).

**Figure 4 materials-14-03119-f004:**
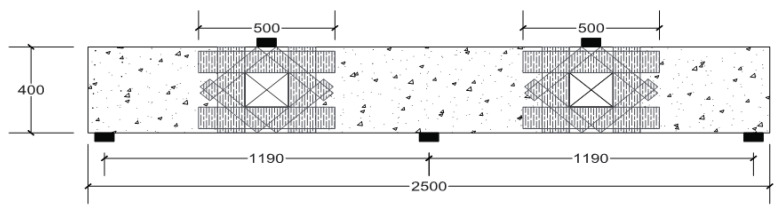
Illustration of the strengthening scheme with CFRP strips around openings of specimen CDB–O–S (all dimensions are in mm).

**Figure 5 materials-14-03119-f005:**
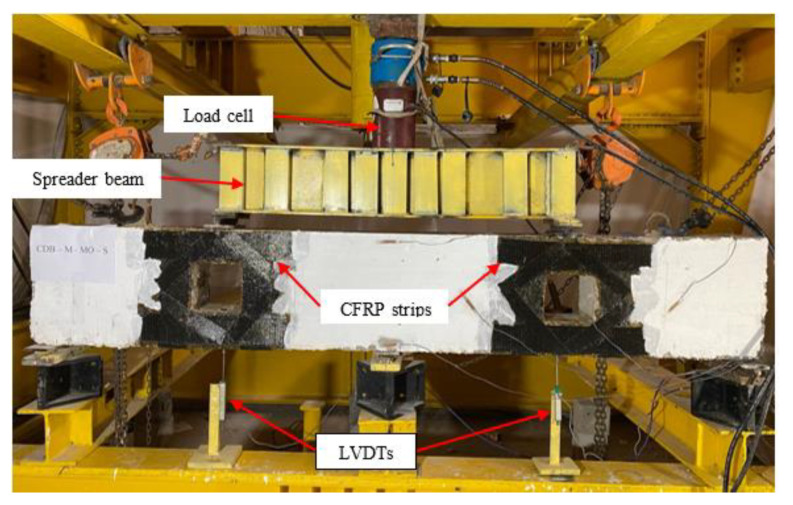
Test setup.

**Figure 6 materials-14-03119-f006:**
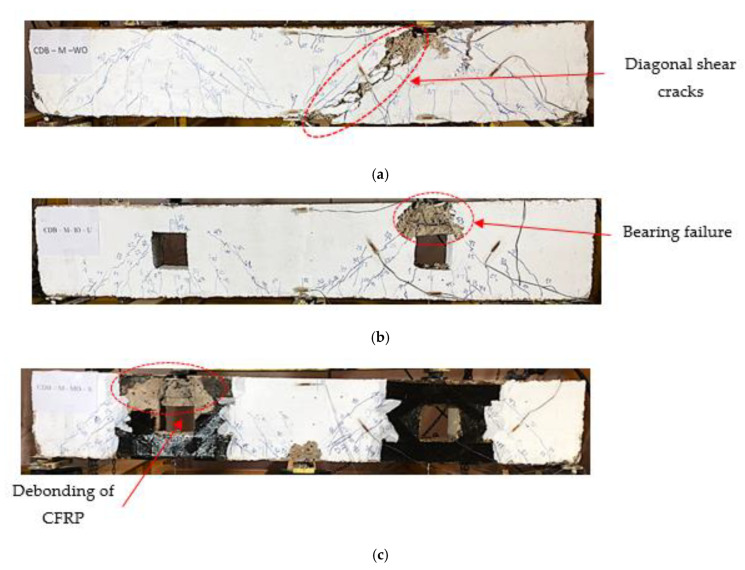
Modes of failure for the tested specimens. (**a**) Specimen CDB–Solid; (**b**) Specimen CDB–O–U; (**c**) Specimen CDB–O–S.

**Figure 7 materials-14-03119-f007:**
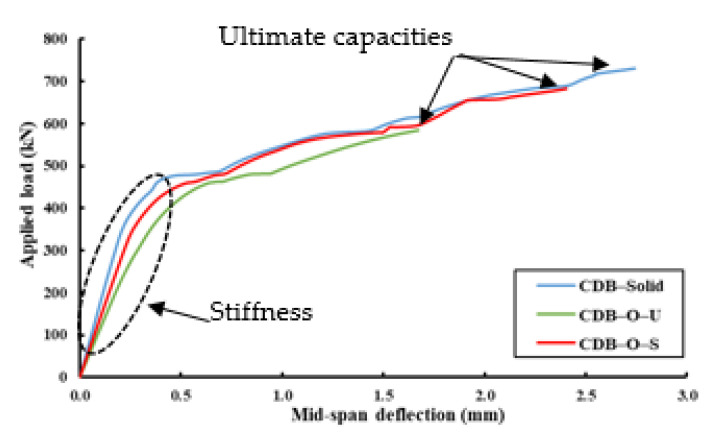
Load mid-span deflection curves for the tested specimens.

**Figure 8 materials-14-03119-f008:**
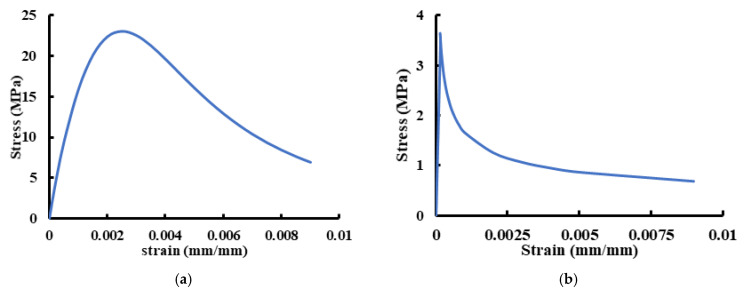
Concrete uniaxial compressive and tensile stress–strain curves. (**a**) In compression; (**b**) in tension.

**Figure 9 materials-14-03119-f009:**
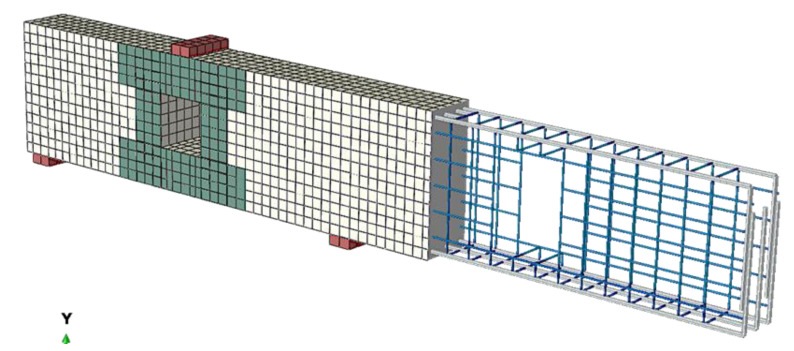
FE mesh and discretization.

**Figure 10 materials-14-03119-f010:**
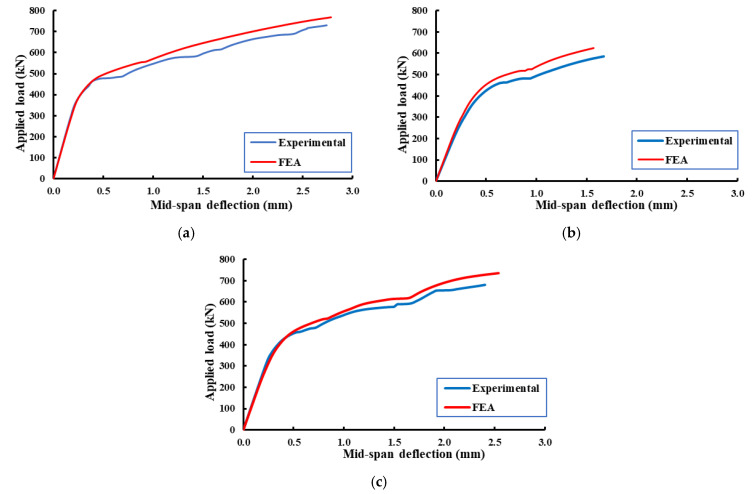
Load mid-span deflection curves for the experimental and FE results. (**a**) Specimen CDB–Solid; (**b**) specimen CDB–O–U; (**c**) specimen CDB–O–S.

**Figure 11 materials-14-03119-f011:**
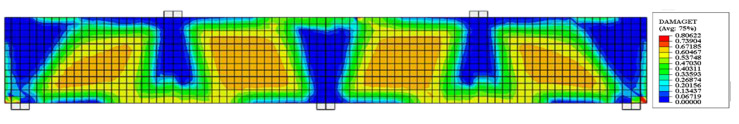
Crack patterns of the tested specimens in experimental setup and ABAQUS. (**a**) Specimen CDB–Solid; (**b**) specimen CDB–O–U; (**c**) specimen CDB–O–S.

**Figure 12 materials-14-03119-f012:**
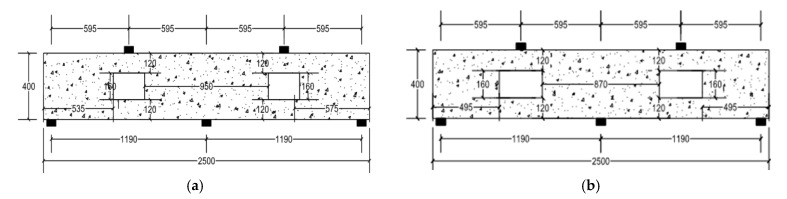
Typical layouts of the RCCDBs with different ratios of the opening dimensions (all dimensions are in mm): (**a**) ratio: 1.5; (**b**) ratio: 2.0.

**Figure 13 materials-14-03119-f013:**
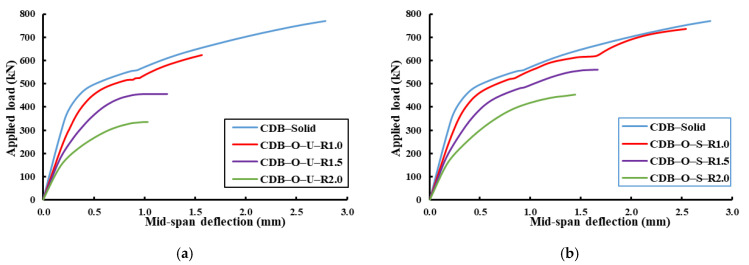
Effect of the ratio of the opening dimensions on the behavior of RCCDBs. (**a**) Un-strengthened beams; (**b**) strengthened beams.

**Figure 14 materials-14-03119-f014:**
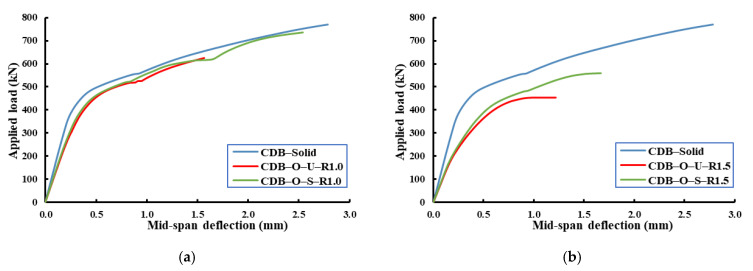
Effectiveness of CFRP strips on strengthening RCCDBs with different ratios of the opening dimensions. (**a**) Opening ratio of 1.0; (**b**) opening ratio of 1.5; (**c**) opening ratio of 2.

**Figure 15 materials-14-03119-f015:**
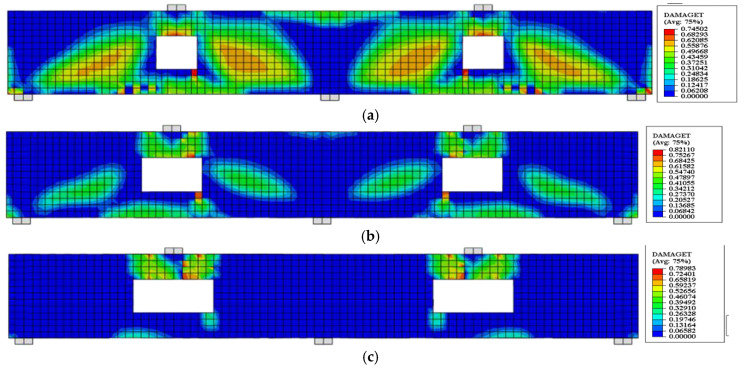
Effect of ratios of the opening dimensions on the FE crack patterns for the un-strengthened RCCDBs. (**a**) CDB–O–U-R1.0; (**b**) CDB–O–U-R1.5; (**c**) CDB–O–U-R2.0.

**Figure 16 materials-14-03119-f016:**
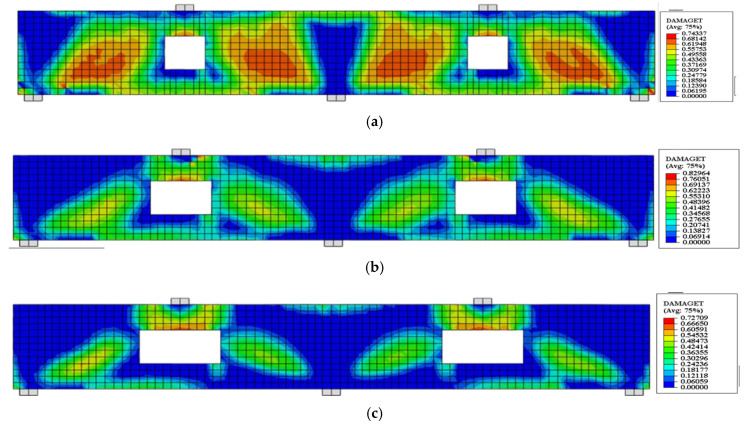
Effect of ratios of the opening dimensions on the FE crack pattern for the strengthened RCCDBs. (**a**) CDB–O–S-R1.0; (**b**) CDB–O–S-R1.5; (**c**) CDB–O–S-R2.0.

**Figure 17 materials-14-03119-f017:**
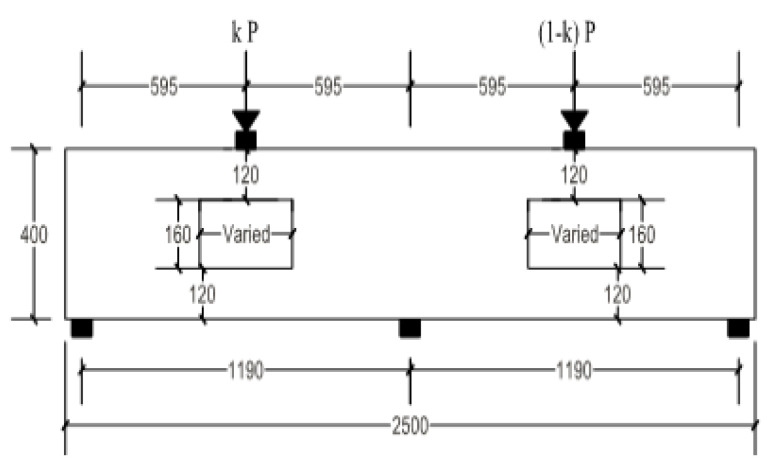
Typical layout of the RCCDBs with openings to consider the load distribution factor.

**Figure 18 materials-14-03119-f018:**
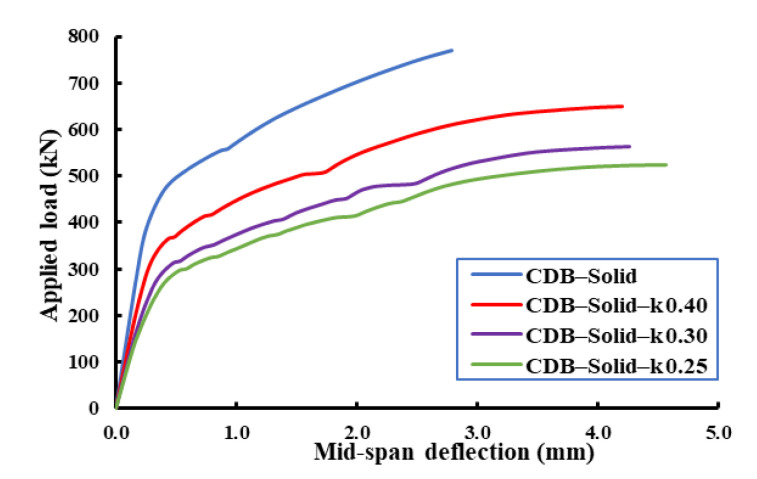
Effect of the load distribution factor on load–mid-span-deflection relationships of RCCDBs without openings (solid).

**Figure 19 materials-14-03119-f019:**
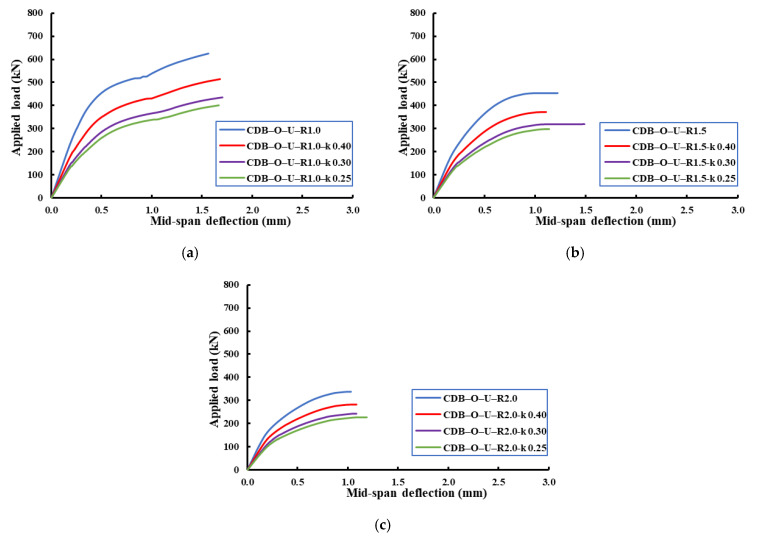
Effect of the load distribution factor on load–mid-span-deflection relationships of the un-strengthened RCCDBs with different ratios of the opening dimensions. (**a**) Opening ratio of 1.0; (**b**) opening ratio of 1.5; (**c**) opening ratio of 2.0.

**Figure 20 materials-14-03119-f020:**
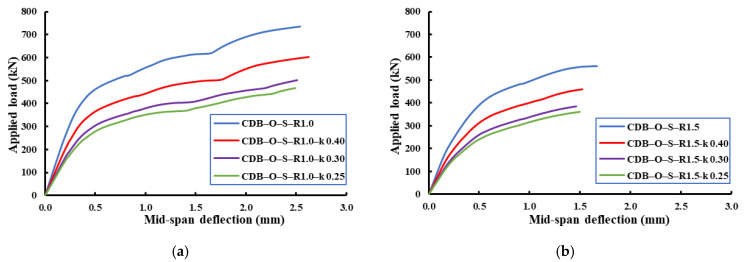
Effect of the load distribution factor on load–mid-span-deflection relationships of the strengthened RCCDBs with different ratios of the opening dimensions. (**a**) Opening ratio of 1.0; (**b**) opening ratio of 1.5; (**c**) opening ratio of 2.0.

**Table 1 materials-14-03119-t001:** Details of the tested specimens.

Specimen *	Cross-Section (mm^2^)	Span (mm)	Openings (mm^2^)	Strengthening
CDB–Solid	160 × 400	1190	Solid	-
CDB–O–U	160 × 400	1190	160 × 160	-
CDB–O–S	160 × 400	1190	160 × 160	CFRP

* CDB: continuous deep beam, O: openings, U: un-strengthened, and S: strengthened.

**Table 2 materials-14-03119-t002:** Mechanical properties of CFRP strips (provided by Sika, Baghdad, Iraq).

CFRP Strips	SikaWrap 231C
Fiber Orientation	0°
Areal weight	450 g/m^2^
Fabric design thickness	0.255 mm (based on the total area of carbon fibers)
Tensile strength of fibers	4800 MPa
Tensile E—modulus of fibers	230,000 MPa
Elongation at break	2.1%
Fabric width	500 mm

**Table 3 materials-14-03119-t003:** Mechanical properties of epoxy adhesive (provided by Sika, Baghdad, Iraq).

Epoxy	Sikadur-330
Appearance	Yellow
Density	1.1 g/cm^3^
Mixing ratio	2:1 by weight (at +25 °C)
Open time	30 min
Tensile strength	25 MPa
E-modulus (Flexural)	2700 MPa
Elongation	3%

**Table 4 materials-14-03119-t004:** Summary of the experimental results.

Specimen	Initial Crack Load (kN)	Ultimate Capacity (kN)	Ultimate Deflection (mm)	Initial Stiffness (kN/mm)	Mode of Failure
CDB–Solid	200	730	2.4	1745	Shear failure
CDB–O–U	140	580	1.7	1142	Bearing failure
CDB–O–S	210	680	2.4	1390	Bearing failure

**Table 5 materials-14-03119-t005:** Parameters of the CDP model used in ABAQUS.

Parameter	Value
φ	33°
ε	0.1
**$f_{bo}/f_{co}$**	29/25
K	2/3
μ	0.0001

**Table 6 materials-14-03119-t006:** Comparisons between the experimental and FE results.

Specimen	Ultimate Capacity (kN)		% Change *	Ultimate Deflection (mm)		% Change *
	Exp.	FE		Exp.	FE	
CDB–Solid	730	770	+ 5.5	2.5	2.8	+ 12
CDB–O–U	580	625	+ 7.7	1.7	1.6	− 5.9
CDB–O–S	680	735	+ 8.1	2.4	2.5	+ 4.2

* % change is calculated based on the experimental results.

**Table 7 materials-14-03119-t007:** Ultimate capacities and mid-span deflections of RCCDBs with different ratios of the opening dimensions.

Specimen	Ultimate Capacity (kN)	% Change *	Ultimate Deflection (mm)	% Change *
CDB–O–U– R 1.0	625	18	1.6	7
CDB–O–S– R 1.0	735		2.5	
CDB–O–U– R 1.5	455	23	1.2	23
CDB–O–S– R 1.5	560		1.7	
CDB–O–U– R 2.0	337	35	1.0	40
CDB–O–S– R 2.0	454		1.4	

* % change is calculated based on the un-strengthened beam.

**Table 8 materials-14-03119-t008:** Effects of the load distribution factor on the FE ultimate capacities and mid-span deflections.

Specimen	Opening Ratio	Distribution Load Factor (k)	Ultimate Load (kN)	Ultimate Mid-Span Deflection (mm)
CDB–Solid	-	0.50	770	2.788
CDB–Solid– k 0.40		0.40	650	4.210
CDB–Solid– k 0.30		0.30	562	4.272
CDB–Solid– k 0.25		0.25	523	4.567
CDB–O–U–R1.0	1.0	0.50	625	1.569
CDB–O–U–R1.0– k 0.40		0.40	513	1.682
CDB–O–U–R1.0– k 0.30		0.30	435	1.704
CDB–O–U–R1.0– k 0.25		0.25	402	1.667
CDB–O–U–R1.5 k 0.50	1.5	0.50	455	2.221
CDB–O–U–R1.5– k 0.40		0.40	373	1.110
CDB–O–U–R1.5– k 0.30		0.30	319	1.486
CDB–O–U–R1.5– k 0.25		0.25	297	1.141
CDB–O–U–R2.0	2.0	0.50	337	1.031
CDB–O–U–R2.0– k 0.40		0.40	281	1.089
CDB–O–U–R2.0– k 0.30		0.30	242	1.085
CDB–O–U–R2.0– k 0.25		0.25	226	1.189
CDB–O–S–R1.0	1.0	0.50	735	2.541
CDB–O–S –R1.0– k 0.40		0.40	603	2.630
CDB–O–S –R1.0– k 0.30		0.30	501	2.510
CDB–O–S –R1.0– k 0.25		0.25	468	2.493
CDB–O–S–R1.5	1.5	0.50	560	1.669
CDB–O–S–R1.5– k 0.40		0.40	459	1.523
CDB–O–S–R1.5– k 0.30		0.30	385	1.461
CDB–O–S–R1.5– k 0.25		0.25	360	1.500
CDB–O–S–R2.0	2.0	0.50	454	1.446
CDB–O–S–R2.0– k 0.40		0.40	366	1.294
CDB–O–S–R2.0– k 0.30		0.30	312	1.327
CDB–O–S–R2.0– k 0.25		0.25	292	1.347

## Data Availability

The data presented in this study are available on request from the corresponding author.

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
