# Peer review of "Strengthening of Continuous Reinforced Concrete Deep Beams with Large Openings Using CFRP Strips"

_materials, 2021, doi:10.3390/ma14113119_

Round 1

Reviewer 1 Report

The submitted research article treats the performance of deep beams investigated experimentally and numerically. Deep beams in construction are very popular due to the architectural constraints for covering wide spans or bear false columns. With different examples, the authors have shown the role of openings in the deep beams and the role of retrofitting by means of CFRP strips.

The manuscript is well-conceived and written with comprehensive and clear language. In my opinion, the manuscript should have the attention of the editorial board as it presents a relevant contribution for the CFRP retrofitting of RC bdeep beams.

The methodology followed by the authors is correct. They have used a standard test to validate a numerical model, which is achieved at an acceptable level. Then different simulations have been conducted, changing some parameters like the opening ratio and the asymmetry of the applied load.

I would highlight to check any error in the manuscript and clarify more into detail some aspects:

  • The referenced literature does not adequately cover state of art. However, it helps to highlight the originality and the author's contribution.
  • Correct the caption of table 2 and 3.
  • Correct the captions of figures and tables, make them more explicative.
  • Clarify the role of mesh and adopted model in the difference between the experimentaly obtained stiffness with the numerically simulated one.
  • The authors should better clarify the role of retrofitting and the retrofitting pattern use. Its correlation with the practical cases. Compare better the damage pattern with openings with the imposed retrofitting.
  • Avoid highlighting trivial conclusions like the role of opening in the stiffness and the capacity.
  • Highlight more the role of retrofitting and try to draw estract of the conclusions from the conducted analyses.
  • Reinforcement and the design capacity plays a paramount role in the damage patterns and load bearing capacity. As in all retrofitting, it is a crucial starting point for evaluating the effectiness of the retrofitting itself. A better treatment in this regrd may be beneficial.

Reviewer 2 Report

The authors investigated the behavior of reinforced concrete continuous deep beams strengthened by CFRP strips. The research is interesting but they should answer and clarify the following comments. After major revision, the paper can be considered for publication.

1- The main contribution of the current study should be clarified. The main highlights of the research should be discussed. 

2- The literature review of the article is really poor. It should be improved by considering some recent and relevant papers in this subject.

3. The main differences between the present research and the following reference should be clarified:

Materials 2020, 13, 2804; doi:10.3390/ma13122804

4- The conclusion part should be re-written. This part should include the main and original results obtained by the present research. 

5- The numerical modelling in ABAQUS should be more explained in detail. 

6- More explanations should be provided for the failure modes of the RCCDBs. 

Reviewer 3 Report

This paper presents an experimentally and numerically investigation on the effectiveness of using carbon fibre reinforced polymer (CFRP) strips as a strengthening technique to externally strengthen reinforced concrete continuous deep beams (RCCDBs) with large openings. The paper is interesting and involves both numerical and experimental data.
The topic is worthy of investigation, and it is relevant for this journal.
General Comments:
1) The introduction section can be improved
2) The description of the experimental program is well described. 
3) The finite element model is well described but can be improved.   
4) Conclusion Section needs to be improved.
Recommendations and Queries:
1) I suggest improving the literature review about the numerical models that focused on different aspects of the FRP system as a retrofitting system for concrete beam:
10.3390/FIB8060042
2) I also suggest improving the discussion by introducing another kind of retrofitting systems like SRG or FRCM:
10.3390/jcs4040182
3) The description of the CDP must be improved with the low used for the damage parameters, please see the references at points one and two.
4) The boundary conditions and the mesh adopted is well described. 
5) The results in terms of loading displacement curves are good. Instead, the damage pattern is actually smeared. It is quite strange. But I suppose it is because you made an erroneous evaluation of the penalty parameter dt.
6) Did you test a control beam? 

Reviewer 4 Report

The manuscript Strengthening of Continuous Reinforced Concrete Deep Beams 2 with Large Openings Using CFRP Strips studies the effect of carbon fiber reinforced polymer (CFRP) strips on the mechanical behavior of concrete deep beam with large opening. The experimental and simulation results are reported in detail and well presented. In general, the review thinks the manuscript is suitable for publication. However, the following comments should be considered.

  1. The CDB-O-S sample is compared to samples without opening and reinforced strips. Should a sample with reinforced strips but without opening be considered as a control sample?
  2. The comparison between different sample is presented in percentage. For instance, line 23-27, “21% and 7% decrease” and “17% increase”. As a non-expert, the review finds these numbers hard to access. What is the allowance for the strength and mid-span deflection for these beams in practice? How will it compare to other kinds of reinforced concrete?
  3. Table 8 presented a lot of data. However, it hasn’t been summarized and discussed in the text.

Round 2

Reviewer 2 Report

The reviewer thanks the authors for considering all comments and answer to all questions. 

Reviewer 3 Report

The manuscript can be published in the present form